# Bi-Level Motion Imitation for Humanoid Robots

**Wenshuai Zhao[1], Yi Zhao[1], Joni Pajarinen[1], Michael Muehlebach[2]**
[1] Aalto University, Finland
[2] Max Planck Institute for Intelligent Systems, Germany

**Abstract:** Imitation learning from human motion capture (MoCap) data provides a promising way to train humanoid robots. However, due to differences in morphology, such as varying degrees of joint freedom and force limits, exact replication of human behaviors may not be feasible for humanoid robots. Consequently, incorporating physically infeasible MoCap data in training datasets can adversely affect the performance of the robot policy. To address this issue, we propose a bi-level optimization-based imitation learning framework that alternates between optimizing both the robot policy and the target MoCap data. Specifically, we first develop a generative latent dynamics model using a novel self-consistent auto-encoder, which learns sparse and structured motion representations while capturing desired motion patterns in the dataset. The dynamics model is then utilized to generate reference motions while the latent representation regularizes the bi-level motion imitation process. Simulations conducted with a realistic model of a humanoid robot demonstrate that our method enhances the robot policy by modifying reference motions to be physically consistent[1].

**Keywords:** Humanoid Robots, Imitation Learning, Latent Dynamics Model

## 1 Introduction

The use of human motion capture (MoCap) data as reference trajectories offers a promising way to design powerful humanoid robot controllers [1, 2, 3, 4]. After appropriate motion retargeting these close-expert reference trajectories can be directly imitated by robots, reducing the need for extensive reward engineering typically required in reinforcement learning [5, 3]. Existing motion imitation works either learn the motion styles in a generative adversarial way [6, 2, 7, 3] or directly learn to track the provided motion trajectories [1, 8]. While the former method, based on generative adversarial imitation learning (GAIL) [9], avoids the exact definition of similarity between reference motions and robot trajectories, its min-max computational formulation usually suffers from unstable learning and sample inefficiency [10, 11]. The latter method, however, can also be problematic because the reference motion is often noisy and physically infeasible for realistic humanoid robots due to embodiment differences such as different force and joint limits between humans and robots [4]. Consequently, including such data may degenerate the policy learning of the robot [4].

The aforementioned issues arising from noisy and physically infeasible reference motion have been mainly studied in the field of motion retargeting [12, 13, 14]. For example, in order to create natural motions for various animated characters, researchers pursue retargeting the human MoCap motions into physically consistent motions of new characters, which in our case corresponds to humanoid robots. The common approach used in physics-based retargeting hinges on trajectory optimization with known dynamics of the target robot and constraints that arise from the reference trajectories [14, 15]. However, the resulting optimization problem is often complex and includes specific domain knowledge. There is therefore an emergent need for a learning-based method that does not rely on an explicit dynamics model while guaranteeing physical consistency at the same time. We address this need by proposing the Bi-Level Motion Imitation (BMI) framework.

---

[1] Project website: https://sites.google.com/view/bmi-corl2024.

8th Conference on Robot Learning (CoRL 2024), Munich, Germany.

Our method shares a similar bi-level optimization idea with differential optimal control [15] but does not need a prior dynamics model and human-specified constraints. Specifically, BMI first learns a generative latent dynamics model based on a novel self-consistent generative auto-encoder (SCAE) from the reference motions. SCAE regularizes normal auto-encoder training with a latent reconstruction error and captures the essential motion patterns with sparse and well-structured latent representations. This enables us to sample latent parameters and reconstruct new motions, which are used to train the humanoid robot policy (*pre-training* step). After pre-training, BMI further finetunes both the decoder and the robot policy as a bi-level optimization problem. In this way, the decoder learns to return reference motions that are physically consistent. At the same time, the robot further improves its policy by imitating updated reference motions. We constrain the decoder updates to ensure that the reconstructed motions stay close to the original motions in the latent space, which prevents the decoder from degenerating into trivial motions that are far from the desired motion patterns in the human MoCap data.

We evaluate BMI on the MIT Humanoid Robot [16] in simulation, where we imitate motions from human MoCap data. The experiments first show that the proposed SCAE-based latent dynamics model learns structured motion representations. In the subsequent pre-training, the improved latent representation learned by SCAE also enhances policy learning compared to the baseline latent dynamics model. Finally, our bi-level fine-tuning with latent space regularization updates the decoder to construct reference motions that are physically consistent for the robot and retain the original patterns at the same time. Our experiments show that the robot policy can be further improved by imitating the updated motions.

The key contributions of this paper can be summarized as follows: (i) We propose a self-consistent latent dynamics model that is able to learn sparse and structured representations for human motions. (ii) We propose a bi-level motion imitation framework to update the decoder and the robot policy at the same time, which enhances the generated motions with physical consistency and closeness to the original human MoCap trajectories. (iii) We evaluate our method on a humanoid robot and imitate up to 13 different motions with a single policy. The experiments highlight improved policy learning with the proposed latent dynamics model and bi-level motion imitation framework.

## 2    Related Work

We first discuss existing reference-based humanoid imitation learning methods. Methods addressing the problem of physically inconsistent reference motions are discussed subsequently.

**Humanoid Motion Imitation**    Imitating from human MoCap data is an efficient way for humanoid robots to learn agile and natural-looking skills [1]. Recent works [7, 17] based on generative adversarial imitation learning (GAIL) [9, 2] in animation have succeeded in training humanoid robots to track various human motions using a large MoCap dataset such as AMASS [18]. Nonetheless, the success may be partially attributed to the unrealistic humanoid robot that is used. With up to 69 DoFs, unlimited force, and even assistive external forces [19], the simulated robot is massively overactuated and can, in principle, perfectly track the given reference motions. It is therefore unclear whether the approaches in animation [7, 17] can be transferred to more realistic robots. As the reference motions can be physically infeasible for robots, including them in the training dataset can result in sub-optimal mimicking behaviors or even complete failure in imitation [13]. The authors from [20] train whole-body humanoid controllers that only replicate upper-body movements while the lower body is restricted to track a given forward velocity for the base. An alternation has been proposed in [4] where the infeasible motions are explicitly removed by a privileged simulated imitator. Fourier Latent Dynamics (FLD) [8] employs a fallback mechanism to replace the given reference motions with default motions when the reference is far from the training motions.

**Physically Consistent Motion Retargeting**    Motion retargeting describes the process of mapping the human MoCap data to target robot configurations such that downstream motion imitation can be performed. While common motion retargeting methods [21, 13] such as inverse kinematics-based

methods can generate visually convincing motions, these motions could be physically infeasible for humanoid robots. In order to obtain physically consistent motion retargeting, existing methods are usually formulated as trajectory optimization problems constrained by robot dynamics [22, 12, 14, 15]. For instance, differential optimal control [15] alternatively optimizes the retargeting parameters with manually defined contact constraints and the robot trajectories based on the retargeting as a bi-level optimization problem. However, it is often tedious to model the complex robot dynamics and these methods are therefore hard to generalize across different robots. In contrast, our method is purely data-driven.

## 3 Preliminaries

Our method involves modifying a latent dynamics model, which maps the motions through an auto-encoder [23] into latent space and back, in order to generate motions for the robot that are physically consistent and at the same time close to the desired motion patterns in the original MoCap dataset. However, measuring the closeness between the original trajectory and the generated physically-consistent reference motion for the robot, is challenging [24]. We address this problem by introducing a structured motion representation and incentivizing closeness in the latent space. Our proposed latent dynamics model is inspired by FLD [8], a structured motion representation method that explicitly enforces the periodicity of motions in the latent space by transforming the learned latent representation into the frequency domain [25].

The structure of FLD is illustrated in Figure 7 in the appendix. We denote a given trajectory segment of length $H$ in $d$-dimensional state space by $\tau_t = (s_{t-H+1}, \cdots, s_t) \in \mathbb{R}^{d \times H}$, where $t$ denotes time and $s_t$ the state at time $t$. The trajectory segment $\tau_t$ represents the input to the auto-encoder, where the encoder embeds the original motion trajectory into a latent space with $c$ channels, denoted by $z_t \in \mathbb{R}^{c \times H}$. In order to explicitly account for the periodicity of the motions, FLD builds on earlier work on Periodic Autoencoders (PAEs) [25] and includes a differentiable Fast Fourier Transform (FFT) layer. The FFT layer returns the frequency $f_t$, amplitude $a_t$, and offset $b_t$ of the latent motion embeddings, while a separate phase $\phi_t$ is computed by an additional fully connected (FC) layer and an atan2 operation. This transformation is denoted as $p$:

$$z_t = \text{enc}(\tau_t), \qquad (\phi_t, f_t, a_t, b_t) = p(z_t), \tag{1}$$

where $\phi_t, f_t, a_t, b_t \in \mathbb{R}^c$ and enc is the encoder. Particularly, FLD improves PAE with a multi-step forward prediction to approximate the subsequent latent vectors by unrolling the latent phase. For a local range of $N$ subsequent trajectory segments $\{\tau_t, \tau_{t+1}, \cdots \tau_{t+N}\}$, we assume that the segments share the same latent parameters $f_t, a_t, b_t$ while differing only in their phases $\phi_{t+i}$. Furthermore, $\phi_{t+i}$ can be approximated by $\phi_{t+i} \approx \phi_t + i f_t \Delta_t$, where $\Delta_t$ denotes the time step. This results in,

$$\hat{z}'_{t+i} = \hat{p}(\phi_t + i f_t \Delta_t, f_t, a_t, b_t), \qquad \hat{\tau}'_{t+i} = \text{dec}(\hat{z}'_{t+i}), \tag{2}$$

where dec is the decoder. We denote by $\hat{p}$ the embedding reconstruction process from the frequency domain,

$$\hat{z}_t = \hat{p}(\phi_t, f_t, a_t, b_t) = a_t \sin(2\pi(f_t \mathcal{T} + \phi_t)) + b_t, \tag{3}$$

where $\mathcal{T}$ represents a known time window with $H$ evenly spaced samples [25]. We note that $\hat{z}'_{t+i}, \hat{s}'_{t+i}$ are different from $\hat{z}_t, \hat{s}_t$ as they are approximated by the multi-step forward prediction from the trajectory $\tau_t$. This motivates the following loss function that is used in FLD,

$$L^N_{\text{FLD}} = \sum_{i=0}^{N} \alpha^i |\hat{\tau}'_{t+i} - \tau_{t+i}|^2, \tag{4}$$

where $\alpha$ is a decay factor and $|\cdot|$ denotes the Euclidean distance.

## 4 Method

The proposed method involves a three-stage training procedure. (i) In the first stage, we learn a generative latent dynamics model from the original MoCap data that has been kinematically retargeted

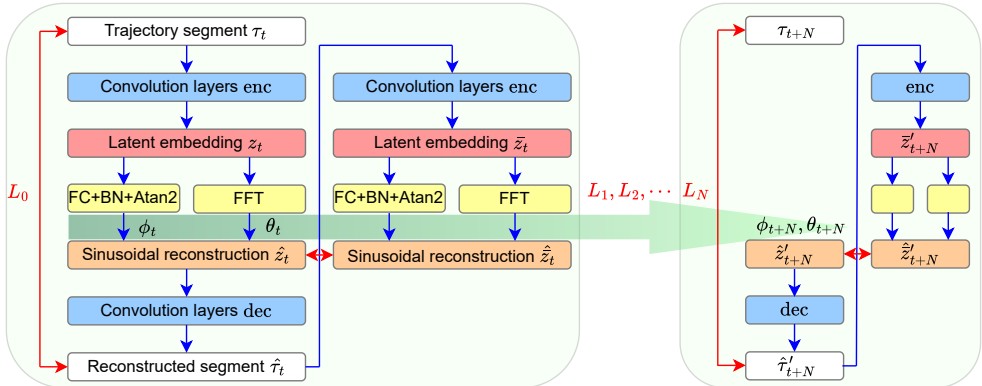

Figure 1: Structure of the proposed self-consistent auto-encoder (SCAE). The encoder enc first encodes the original trajectory $\tau_t$ into latent space $z_t$. Fourier transformation is then applied to $z_t$ to get latent parameters $\theta_t = (f_t, a_t, b_t)$ while a separate MLP module learns $\phi_t$. A sinusoidal function reconstructs the latent embedding $\hat{z}_t$, followed by the decoder dec recovering the input trajectory $\hat{\tau}_t$. Particularly, we re-input $\hat{\tau}_t$ to the encoder to obtain reconstructed latent embedding $\hat{\bar{z}}_t$. Therefore, SCAE consists of both motion and latent reconstruction losses, as indicated by red arrows. We follow FLD to make multi-step predictions and thus the final loss sums $L_0, \cdots, L_N$.

to the humanoid. We introduce a self-consistent auto-encoder trained using both reconstruction error and latent regularization, to capture the desired patterns embedded in the noisy kinematic motions more effectively. (ii) The second stage samples latent parameters encoded by the self-consistent dynamics model and then decodes these latent samples into the state space. The decoded states are used as the reference motions to pre-train the robot policy. (iii) We perform bi-level imitation by fine-tuning the policy and updating the decoder at the same time. Crucially, this bi-level optimization is constrained within the latent space, ensuring that the decoder generates motions that closely adhere to physics-based robot trajectories while preserving the original motion patterns intended for imitation. The following paragraphs explain the three-step procedure in detail.

## 4.1 Self-Consistent Latent Dynamics

Although FLD learns structured latent representations and shows accurate reconstruction, we find that the decoded motions with small reconstruction errors are not guaranteed to stay close to the original motions in the latent space. This means that the learned latent representation overfits to current data and is not robust to noise in the motions. In contrast, with our bi-level motion imitation framework, we introduce a latent representation that focuses on the general motion patterns instead of nuances and noise. This is important, since the nuances are likely to change when converted to be physically consistent in the fine-tuning step.

We address the above gap by a Self-Consistent Auto-Encoder (SCAE). Specifically, we propose to regularize FLD learning with a latent reconstruction error. A similar idea has been applied to VAE [26] but has not been investigated in deterministic auto-encoders for motion generation. Figure 1 shows the structure of SCAE, where the reconstructed trajectory $\hat{\tau}_t$ is fed into the encoder again in order to obtain a reconstructed latent representation $\hat{\bar{z}}_t$ from the decoded motion $\hat{\tau}_t$. We retain the multi-step prediction in FLD and thus our SCAE training loss is

$$L_{\text{SCAE}}^N = \sum_{i=0}^{N} \alpha^i (|\hat{\tau}'_{t+i} - \tau_{t+i}|^2 + \beta|\hat{\bar{z}}'_{t+i} - \hat{z}'_{t+i}|^2), \tag{5}$$

where $\beta$ is the coefficient of the latent reconstruction error and where we evaluate the loss on the entire dataset. The reconstructed latent representation $\hat{\bar{z}}'_{t+i}$ is computed by feeding the reconstructed trajectory $\hat{\tau}'_{t+i}$ into the encoder, the Fourier transform layer and the sinusoidal reconstruction layer. Note that $\hat{\tau}'_{t+i}$ is obtained by the multi-step forward prediction in Equation 2.

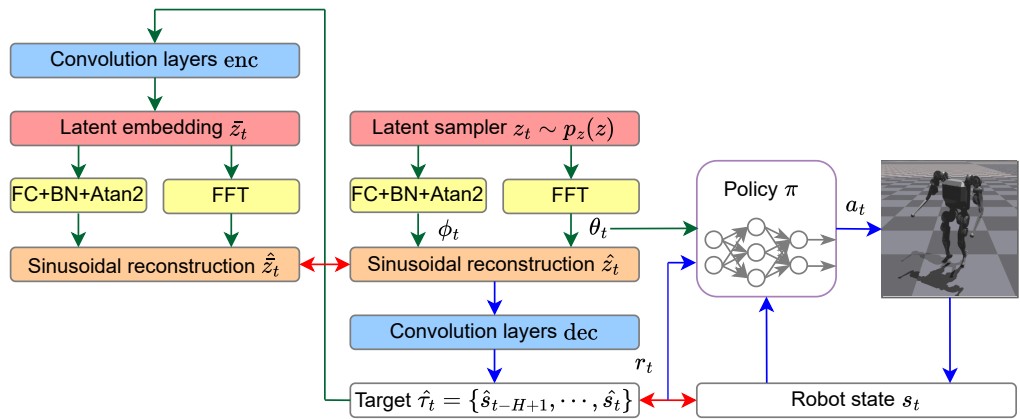

Figure 2: Bi-level motion fine-tuning (BMI) optimizes both the robot policy and the decoder alternatively. The learning begins by sampling from the learned latent space $p(z)$ and decoding these latent samples into target reference motions for robot imitation. The decoder's loss function comprises two components, as indicated by the red arrows: (1) the mean squared error (MSE) between the robot's trajectory and the decoded trajectory, and (2) the latent reconstruction error between the sampled latent embeddings $\hat{z}_t$ and the embeddings of the decoded trajectories $\hat{\bar{z}}_t$.

With a perfect decoder, the reconstructed motion $\hat{\tau}_t$ is exactly the same as the original motion $\tau_t$ leading to zero *latent reconstruction error* $|\hat{\bar{z}}'_{t+i} - \hat{z}'_{t+i}|^2$. However, this is usually not achievable. Although $|\hat{\bar{z}}'_{t+i} - \hat{z}'_{t+i}|^2$ generally decreases as the decoder learns to reconstruct the trajectory, our experiments show that $|\hat{\bar{z}}'_{t+i} - \hat{z}'_{t+i}|^2$ is not minimized when only optimizing the *motion reconstruction error* $|\hat{\tau}'_{t+i} - \tau_{t+i}|^2$. In contrast, due to the latent reconstruction regularization, SCAE enforces the learned latent representation to be consistent with its decoded motions.

## 4.2 Pre-Training Policy

In this stage, we train our robot policy to track the given reference motions regardless of the feasibility of these motions as done in existing motion imitation works [1, 4]. In contrast to directly sampling trajectories from the original motion dataset to train the robot policy, we sample from the latent space of the SCAE and inform the robot policy with the sampled latent parameters as the target motion information. The self-consistent latent dynamics model provides two advantages compared to using the original datasets. (i) We can interpolate latent parameters to generate motion transitions and new motions, as discussed in FLD [8] and PAE [25]; (ii) We observe that a learned latent representation as the tracking goal for the robot is more concise with essential motion patterns and focuses less on motion nuances, which is beneficial for policy learning.

The policy pre-training procedure is illustrated in Figure 2 without the green arrow modules (these are only used in the next fine-tuning stage). For each episode, we sample a set of latent variables $z_t$ from the pre-collected buffer $p_z(z)$ during SCAE training. We then obtain $(\phi_t, f_t, a_t, b_t) = p(z_t)$ by the following FC and FFT layers. Note that instead of taking the learned phase $\phi_t$, we uniformly sample an initial phase variable $\phi_0 \in \mathbb{R}^c$ from a fixed range and update $\phi_t$ according to the latent dynamics in Equation 2,

$$\phi_t = \phi_{t-1} + f_{t-1}\Delta t, \qquad \{f_t, a_t, b_t\} = \theta_t = \theta_{t-1}. \tag{6}$$

We maintain the same frequency $f_t$, amplitude $a_t$, and offset $b_t$ for the episode. The latent variables are then used to reconstruct a motion trajectory

$$\{\hat{s}_{t-H+1}, \cdots, \hat{s}_t\} = \hat{\tau}_t = \text{dec}(\hat{p}(f_t, a_t, b_t, \phi_t)), \tag{7}$$

where the most recent state $\hat{s}_t$ serves as the target state to compute the robot tracking reward at the current timestep. The policy is learned using proximal policy optimization [27].

### 4.3  Bi-Level Fine-Tuning

This step ensures physical consistency of the reference motions generated by the decoder. Obtaining reference motions that are physically consistent is important as it facilitates policy learning and encourages the robot to learn a versatile set of skills, in particular when the humanoid robots are under-actuated and have restricted torque limits [7, 20, 4]. We propose to convert these unphysical motions into physically consistent ones by a bi-level fine-tuning to maximize the benefit of human MoCap data. This represents an important difference from recent works that address this problem by only tracking upper body movements [20] or filtering out the unlearnable motions [4].

Figure 2 shows the structure of our bi-level fine-tuning. In this stage, we alternatively optimize the policy $\pi$ and the decoder dec while freezing the convolutional encoder enc and the FC, BN layers. In this way, the decoder is encouraged to generate motions close to the robot trajectories, which are physically consistent by design. We further regularize the decoder optimization by constraining the generated motions to be close to the original motions in the latent space. This prevents the decoder from generating trivial motions by simply copying the robot, failing to improve the robot policy further. The bi-level optimization problem is formulated as,

$$\min_{\theta_{\text{dec}}} \mathbb{E}_{z_t \sim p_z(z), s_t \sim \pi_{\theta_\pi^*}}[|\hat{s}_t - s_t|^2 + \beta|\hat{\bar{z}}_t - \hat{z}_t|^2],$$
$$\theta_\pi^* \in \arg\min_{\theta_\pi} \mathbb{E}_{z_t \sim p_z(z), s_t \sim \pi_{\theta_\pi}}[|\hat{s}_t - s_t|^2], \tag{8}$$

where $\theta_{\text{dec}}$ denotes the parameters of the decoder and $\pi_{\theta_\pi}$ the robot policy with parameters $\theta_\pi$. With the proposed regularized bi-level motion imitation, the decoder is updated to generate motions physically consistent with the robot while retaining the desired motion patterns in the dataset. As a result, we observed that the robot further improves the policy during this fine-tuning step.

## 5  Experiments

We evaluate BMI on the MIT humanoid robot [16] in Isaac Gym [28] while keeping the joint and force limits unchanged. We extend the dataset from FLD [8] by including four additional difficult motions, i.e., *jump, kick, spin-kick*, and *cross-over* [1]. In total, we have trajectories from 13 different motions. Our experiments examine both the learned dynamics model and policy performance.

### 5.1  Latent Dynamics Model Learning

**Motion and Latent Reconstruction**   Figure 3b shows that our method and FLD can reconstruct the original motions with comparable accuracy. However, our method with explicit self-consistency constraints achieves significantly lower latent reconstruction error, i.e., $|\hat{\bar{z}}'_{t+i} - \hat{z}'_{t+i}|^2$, as shown in Figure 3a. Therefore, the proposed self-consistent regularization improves the latent reconstruction without sacrificing the motion reconstruction accuracy. An ablation study on the coefficient $\beta$ of latent reconstruction loss in the appendix shows that SCAE is robust to a wide range of $\beta$ values.

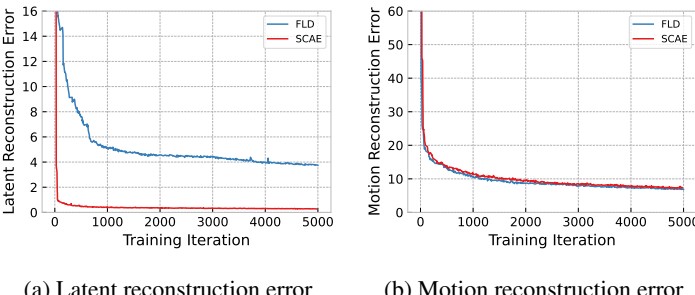

(a) Latent reconstruction error        (b) Motion reconstruction error

Figure 3: Reconstruction error during training: (a) The reconstruction error of latent embeddings. (b) The reconstruction error of the original motion states.

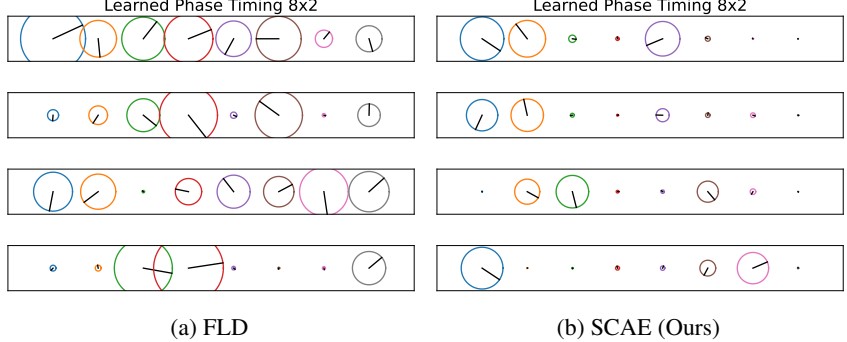

(a) FLD                          (b) SCAE (Ours)

Figure 4: The figure displays the learned latent phases of four motions. Each circle represents a latent channel where the radius is the amplitude and the black bar is the phase timing. Compared to FLD, SCAE takes fewer frequency components and lower amplitudes to represent the same motion.

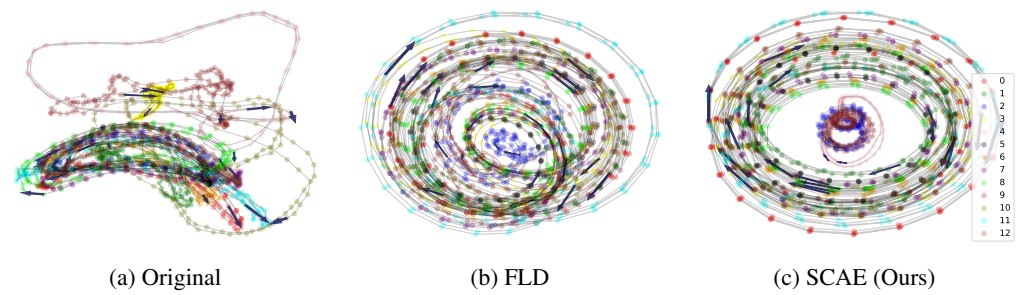

(a) Original                (b) FLD                (c) SCAE (Ours)

Figure 5: The figure shows the latent manifolds for 13 motions. Each color corresponds to a trajectory segment from a motion type. The arrows denote the motion evolution direction. The manifold induced by SCAE shows consistent structures across different motions.

**Learned Latent Manifold**  We visualize the learned latent amplitude $f_t$ and latent phase $\phi_t$ in eight latent channels, computed as in Equation 1, for four motions *run, jog, step fast, jump* in Figure 4, where each row denotes the same motion. Thanks to the latent regularization, our method learns a much sparser representation than FLD, as SCAE takes fewer frequency components to reconstruct the same motions with most channels' amplitudes around zero.

Figure 5 compares the latent structure induced by SCAE with that by FLD, where Figure 5a visualizes the principal components of the original motions. Notably, SCAE demonstrates the most consistent structure across 13 different motions. The circles connecting points with the same color represent the primary period of individual motions and each point denotes a trajectory segment. The radius of a rough circle means that the high-level features throughout a motion can be constant, such as velocity, frequency, etc. The well-shaped latent manifolds learned by SCAE show that our method successfully captures essential motion patterns.

## 5.2   Performance of Policy Learning

We both quantitatively and qualitatively compare the policy learned by FLD, SCAE pre-training, and BMI fine-tuning. Since the target reference motions are noisy and sometimes physically inconsistent for the robot, the commonly used mean square error (MSE) from the reference motions is not an ideal performance metric. Instead, we calculate motion-specific quantities to compare the policy performance. Table 1 and Figure 6b show that BMI achieves the longest kicking time and the most stable standing while kicking. We find, perhaps surprisingly, that without further fine-tuning SCAE improves policy learning in the pre-taining stage and achieves the longest jumping time as shown in Table 1. We hypothesize that the policy improvement in pre-training is due to the change of latent parameterizations used to inform the policy. SCAE learns a sparser representation that makes policy

Table 1: Results on two selected challenging motions: *kick* and *jump*.

| Motion (Metric)\Algo. | FLD | SCAE(Ours) | BMI(Ours) |
|---|---|---|---|
| Kick (Time (%)) | 64.4 | 61.2 | **71.3** |
| Kick (Height (m)) | 0.157 | 0.152 | **0.164** |
| Jump (Time (%)) | 32.5 | **36.2** | 35.2 |

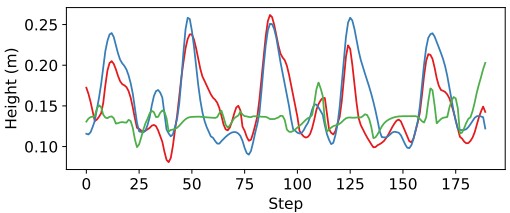

(a) Kicking foot height when kicking

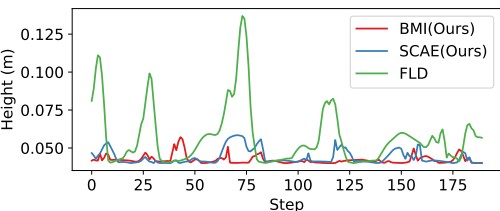

(b) Standing foot height when kicking

Figure 6: Comparison on the challenging *kick* task: The left figure shows the height of the kicking foot during one *kick* trajectory with multiple trials, where both SCAE and BMI outperform FLD in each kick (one mode of the curve). The right figure shows the height of the standing foot where BMI and SCAE are more stable with a lower height of the standing foot.

learning easier. More qualitative results on the other motions can be found on the website. The videos qualitatively show the improvement via BMI on diverse motions such as *cross-over, stride* and *step*. The experiments therefore confirm our hypothesis that by updating the decoder the robot policy can be further improved.

We conduct additional experiments to thoroughly analyze our method, which can be found on our website. (i) Two zero-shot sim-to-sim experiments show that the learned policy works well even when the robot is added a 5kg mass block. (ii) We also visualize the motion changes of the decoded trajectories before and after bi-level fine-tuning. The video displays an increased arm swing in the fine-tuned *stride* motion, suggesting greater physical consistency with the robot's dynamics. (iii) The learned latent dynamics model can potentially function as a generative model to synthesize new motions. By simply interpolating the latent amplitude and frequency, we can generate new motions that are kinematically consistent.

## 6 Limitations & Conclusion

**Limitations**  While the proposed bi-level motion imitation framework alleviates problems arising from physically inconsistent reference motions, the approach relies on a decent robot policy in the pre-training stage. Moreover, since the given references obtained by motion retargeting from human MoCap data are not the optimal targets for robot imitation, the choice of metric to quantify robot tracking performance is an open question. We also note that it would be beneficial to scale up our method to a large-scale MoCap dataset such as AMASS [18] and apply the learned policy to a real-world humanoid robot via sim-to-real techniques [4, 29].

**Conclusion**  This paper presents BMI, a novel bi-level motion imitation framework that minimizes the robot tracking error by alternatively optimizing the robot policy and the motion generation model while being regularized by latent space constraints. Our proposed self-consistent auto-encoder captures the essential motion patterns with sparse and well-structured latent representations, providing a reliable anchor to regularize the decoder to stay close to the desired motion patterns in the dataset. In contrast to existing optimal control methods, BMI addresses the difficulty of including physically inconsistent reference motions in a purely data-driven way and is scalable to large-scale human MoCap datasets. Our experiments on the realistic MIT humanoid robot show that BMI not only improves the pre-trained policy on challenging tasks but also further stabilizes the learned motions.

**Acknowledgments**

The authors thank the support of the German Research Foundation and the Max Planck Institute for Intelligent Systems, Tuebingen (Germany). Wenshuai Zhao, Yi Zhao, and Joni Pajarinen acknowledge funding by the Research Council of Finland (decision numbers 345521, 357301). The authors thank Klaus-Rudolf Kladny for the insightful discussion.

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

# A Appendix

## Contents

## A.1 Structure of FLD

The structure of FLD follows PAE [25] using an auto-encoder to learn a generative dynamics model, where the encoder and the decoder are composed of 1D convolutional layers. In order to enforce the periodicity in the latent manifolds, PAE parameterized each latent channel as a sinusoidal function where the amplitude, frequency, and offset are computed by a differentiable Fast Fourier Transform layer while the phase is determined with a fully connected layer followed by an Atan2 operation. Inspired by the observation that the learned latent frequency, amplitude, and offset by PAE stay nearly constant along the trajectories, FLD improves PAE by combining the structure with a multi-step prediction step as in Equation 4.

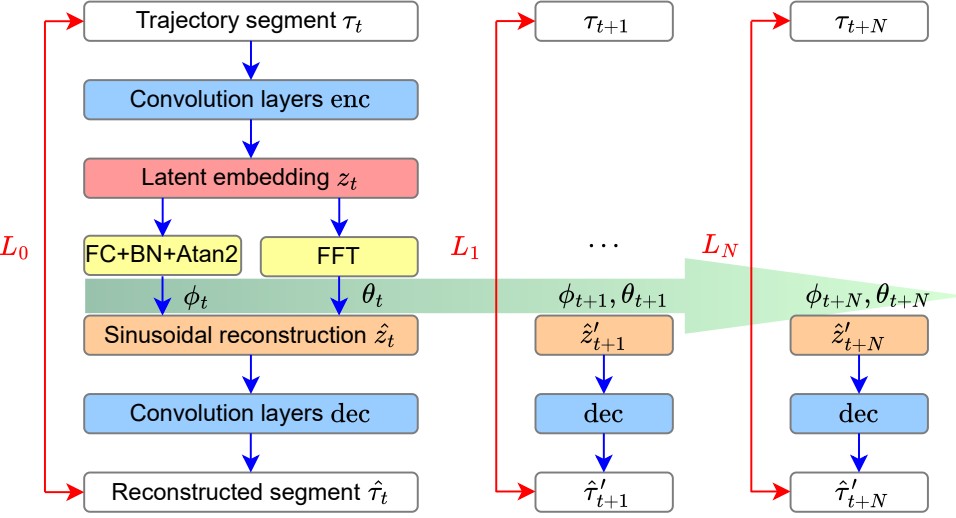

Figure 7: Multi-step forward prediction structure of FLD.

## A.2 Pseudo Code of BMI

Algorithm 1 shows the details of BMI training.

---

**Algorithm 1** Bi-Level Motion Imitation (BMI)

---

**Input:** SCAE encoder enc and decoder dec, latent parameters of the original motions $p_z(z)$, Pre-trained policy $\pi_{\theta_\pi}$ based on SCAE, initial buffer $\mathcal{D}$
**for** $k = 1$ **to** $K$ **do**
    **Policy Learning:**
    **for** $i = 1$ **to** $M_1$ **do**
        Sample latent targets $z_i \sim p_z(z)$
        Extract the target states $\{\hat{s}_{t-H+1}, \cdots, \hat{s}_t\} = \hat{\tau}_t = \mathrm{dec}(\hat{p}(f_t, a_t, b_t, \phi_t))$
        Rollout robot trajectory $\tau_i \sim p(\pi_{\theta_\pi}, \mathrm{dec}, p_z)$
        Collect the trajectory and latent parameter pairs in the buffer $\mathcal{D} = \{(z_i, \tau_i) | i \sim M_1\}$
        Update robot policy $\pi_{\theta_\pi}$ with PPO or another RL algorithm according to the bottom objective in Equation 8
    **end for**
    **Decoder dec Update:**
    **for** $i = 1$ **to** $M_2$ **do**
        Sample latent parameters and robot trajectories from $\mathcal{D}$
        Update the decoder dec according to the upper objective in Equation 8
    **end for**
**end for**

---

## A.3 Experiment Settings

In this section, we provide more detailed experiment settings. We first introduce our dataset and then explain the state and action spaces used in SCAE and the policy. Finally, we list the architectures and hyper-parameters used in both the dynamics model learning and policy learning.

### A.3.1 Dataset

We use the same dataset as FLD [8] which was originally released in DeepMimic [1]. The human MoCap data were manually processed and retargeted to the humanoid robot. Note that even with careful kinematic retargeting the reference motions can be physically inconsistent to the robot dynamics. Our dataset consists of 13 different motions: *run, jog, step fast, jump, spin-kick, back, side left, jog slow, side right, cross-over, kick, stride, step* and each motion has 10 trajectories collected from different demonstrations. In each trajectory of length 240 steps, the demonstrator performs multiple trials of the same action. For example, in one *kick* trajectory, the demonstrator may continuously kick five times as shown in Figure 6a. In total, we have $13 \times 10 \times 240 = 31200$ data points. Each data point corresponds to a state vector of length 52, where the elements are listed in Table 2 (see below).

Table 2: Elements of one data point (step) in the dataset

| Entry | Symbol | Dimensions |
|---|---|---|
| base position | $p_b$ | 0:3 |
| base rotation | $q_b$ | 3:7 |
| base linear velocity | $v$ | 7:10 |
| base angular velocity | $w$ | 10:13 |
| projected gravity | $g$ | 13:16 |
| joint positions | $q$ | 16:34 |
| joint velocity | $\dot{q}$ | 34:52 |

Note that FLD experiments were only run on nine motions: *run, jog, step fast, back, side left, jog slow, side right, stride, step*, referred to as *normal motions*, which present a mild difficulty for

the robot to track. However, our experiments include an additional four motions, *jump, spin-kick, cross-over, kick*, that are significantly more challenging. FLD fails to learn these complex motions satisfactorily without a specifically designed reward function tailored to each individual motion, while our methods show improved performance on the challenging *kick* and *jump* with unchanged reward design.

### A.3.2  State and Action Spaces

In this section, we introduce the state space used in the latent dynamics model and the observation and action spaces for the robot policy.

**State Space of Latent Dynamics Model**   The state space used in the latent dynamics model is composed of the linear and angular velocities of the robot base $v, w$ in the robot frame, measurement of the gravity vector $g$ in the robot frame, and joint positions $q$ as in Table 3. Note that we use the same setting for both FLD and SCAE.

Table 3: Elements of the state space for latent dynamics model

| Entry | Symbol | Dimensions |
|---|---|---|
| base linear velocity | $v$ | 0:3 |
| base angular velocity | $w$ | 3:6 |
| projected gravity | $g$ | 6:9 |
| joint positions | $q$ | 9:27 |

**Observation Space of Robot Policy**   In addition to the state information used in the latent dynamics model, the robot observes extra information such as joint velocities $\dot{q}$ and its last action $a'$. Moreover, we provide the latent parameters to the robot as the target motion information. Therefore, the observation space is shown as Table 4. Note that we apply domain randomization to the policy training including the observation noise, disturbances of the mass, and disturbances arising from pushing as used in FLD [8].

Table 4: Elements of the observation space for robot policy

| Entry | Symbol | Dimensions | Noise level |
|---|---|---|---|
| base linear velocity | $v$ | 0:3 | 0.2 |
| base angular velocity | $w$ | 3:6 | 0.05 |
| projected gravity | $g$ | 6:9 | 0.05 |
| joint positions | $q$ | 9:27 | 0.01 |
| joint velocities | $\dot{q}$ | 27:45 | 0.75 |
| last actions | $a'$ | 45:63 | 0.0 |
| latent phase | $\sin\phi$ | 63:71 | 0.0 |
| latent phase | $\cos\phi$ | 71:79 | 0.0 |
| latent frequency | $f$ | 79:87 | 0.0 |
| latent amplitude | $a$ | 87:95 | 0.0 |
| latent offset | $b$ | 95:103 | 0.0 |

**Action Space of Robot Policy**   The action space of our robot is of $18$ dimensions, which represent the target positions of $18$ joints in the robot. An underlying PD controller [30] is used to compute the torques to drive each joint. The PD gains are set to $(30.0, 5.0)$ for lower body joints and $(40.0, 5.0)$ for upper body joints, respectively.

### A.3.3  SCAE Training

We introduce first the architecture of neural networks used in SCAE, which is the same as FLD. Then we list the hyper-parameters for training the latent dynamics model.

**Architecture of SCAE**  SCAE shares the same architecture as FLD. The architectures of the encoder enc and decoder dec are shown in Table 5. BN denotes batch normalization and ELU represents the exponential linear unit.

Table 5: Architecture of the neural networks used in SCAE

| Network | Layer | Output size | Kernel size | Normalization | Activation |
|---|---|---|---|---|---|
| encoder | Conv1d | 64x51 | 51 | BN | ELU |
|  | Conv1d | 64x51 | 51 | BN | ELU |
|  | Conv1d | 8x51 | 51 | BN | ELU |
| phase encoder | Linear | 8x2 | – | BN | Atan2 |
| decoder | Conv1d | 64x51 | 51 | BN | ELU |
|  | Conv1d | 64x51 | 51 | BN | ELU |
|  | Conv1d | 27x51 | 51 | BN | ELU |

**Hyper-Parameters for SCAE Training**  SCAE uses the same hyper-parameters for training FLD as in Table 6. The extra coefficient of the latent reconstruction regularization used in SCAE, i.e., $\beta$ in Equation 5, is set to 1. Adam is used as the optimizer for training the latent dynamics model.

Table 6: Hyper-parameters of SCAE training

| Parameter | Symbol | Value |
|---|---|---|
| step time seconds | $\Delta t$ | 0.02 |
| max training iterations | – | 5000 |
| learning rate | – | 0.0001 |
| weight decay | – | 0.0005 |
| learning epochs | – | 5 |
| mini-batches | – | 4 |
| latent channels | $c$ | 8 |
| trajectory segment length | $H$ | 51 |
| multi-step prediction length | $N$ | 50 |
| propagation decay | $\alpha$ | 1.0 |

### A.3.4  Policy Training

**Architecture of Policy & Value function**  The neural network architectures of the learning policy $\pi$ and the value function $V$ used in PPO are shown in Table 7.

Table 7: Architecture of the neural networks used in policy training

| Network | Type | Hidden | Output size | Activation |
|---|---|---|---|---|
| policy $\pi$ | MLP | 128, 128, 128 | 18 | ELU |
| value function $V$ | MLP | 128, 128, 128 | 1 | ELU |

**Hyper-Parameters for Policy Training**  We use Adam as the optimizer for the policy and value function with an adaptive learning rate with a KL divergence target of $0.01$. The policy runs at 50 Hz. We parallize $4096$ environments in Isaac Gym to collect samples. The summary of the policy training hyper-parameters can be found in Table 8.

**Reward Function for Policy Training**  The reward function used to train the robot policy consists of two categories $r = r^T + r^R$, where $r^T$ denotes the tracking rewards and $r^R$ represents the regularization rewards. The tracking reward calculates the weighted sum of individual rewards on each dimension bounded in $[0, 1]$ with their weights in Table 9,

$$r^T = w_v r_v + w_w r_w + w_g r_g + w_{q_{\mathrm{leg}}} r_{q_{\mathrm{leg}}} + w_{q_{\mathrm{arm}}} r_{q_{\mathrm{arm}}}. \tag{9}$$

The reward of each dimension is generally formulated as,

$$r_i = e^{-\sigma_i |\hat{d}_i - d_i|^2}, \tag{10}$$

Table 8: Hyper-parameters of policy training

| Parameter | Symbol | Value |
|---|---|---|
| step time seconds | $\Delta t$ | 0.02 |
| max training iterations | – | 3000 |
| max episode time seconds | – | 20 |
| learning rate | – | 0.001 |
| steps per iteration | – | 24 |
| learning epochs | – | 5 |
| mini-batches | – | 4 |
| KL divergence target | – | 0.01 |
| discount factor | $\gamma$ | 0.99 |
| clip range | $\epsilon$ | 0.2 |
| entropy coefficient | – | 0.01 |
| parallel training environments | – | 4096 |

where $i$ denotes the $i_{\text{th}}$ dimension. $d_i$ denotes the target value of this dimension while $\hat{d}_i$ represents the reconstructed value. The variable $\sigma_i$ denotes a temperature factor for each reward and can be found in Table 10.

Table 9: Weights of the tracking rewards

| Weight | $w_v$ | $w_w$ | $w_g$ | $w_{q_{\text{leg}}}$ | $w_{q_{\text{arm}}}$ |
|---|---|---|---|---|---|
| Value | 1.0 | 1.0 | 1.0 | 1.0 | 1.0 |

Table 10: Temperature factors of the tracking rewards

| Weight | $\sigma_v$ | $\sigma_w$ | $\sigma_g$ | $\sigma_{q_{\text{leg}}}$ | $\sigma_{q_{\text{arm}}}$ |
|---|---|---|---|---|---|
| Value | 0.2 | 0.2 | 1.0 | 1.0 | 1.0 |

The regularization reward is formulated as Equation 11, where the weights can be found in Table 11 and each term is detailed as follows:

$$r^R = w_{\text{ar}}r_{\text{ar}} + w_{\text{qa}}r_{\text{qa}} + w_{\text{qT}}r_{\text{qT}}, \tag{11}$$

with action rate

$$r_{\text{ar}} = |a' - a|^2, \tag{12}$$

where $a'$ and $a$ denote the previous and current actions, joint acceleration

$$r_{\text{qa}} = |\frac{\dot{q}' - \dot{q}}{\Delta t}|^2, \tag{13}$$

where $\dot{q}'$ and $\dot{q}$ denote the previous and current joint velocity, $\Delta t$ represents the time step, and with joint torque

$$r_{\text{qT}} = |T|^2, \tag{14}$$

where $T$ stands for torque.

### A.3.5 BMI Training

In the bi-level fine-tuning process, we retain most of the hyperparameters from the pre-training stage. Notable exceptions include the following: (i) We set a lower learning rate for the decoder update compared to the rate used in SCAE training. (ii) To align the magnitudes of the latent reconstruction loss and the motion reconstruction loss in Equation 8, we increase the coefficient $\beta$ to 200. The key hyper-parameters used in BMI are summarized in Table 12. These hyper-parameters may be further tuned for improved results. As this is an initial study of bi-level fine-tuning, we tested only a limited number of hyper-parameter configurations in our experiments.

Table 11: Weights of the regularization rewards

| Weight | $w_{ar}$ | $w_{qa}$ | $w_{qT}$ |
|---|---|---|---|
| Value | $-0.01$ | $-2.5 \times 10^{-7}$ | $-1.0 \times 10^{-5}$ |

Table 12: Hyper-parameters of BMI fine-tuning

| Parameter | Symbol | Value |
|---|---|---|
| coefficient of latent reconstruction loss | $\beta$ | 200 |
| learning rate for decoder | – | 0.00001 |
| number of mini-batch for decoder | – | 2 |
| max training iteration | – | 50 |
| epochs for decoder | – | 1 |
| steps per iteration | – | 24 |
| parallel training environments | – | 4096 |

### A.4 More Experiment Results

We show more experiment results in this section, including experiments for both the latent dynamics model learning and the policy learning.

#### A.4.1 Ablation Study on Latent Reconstruction Error

We test a range of $\beta$ values for SCAE training. The results in Figure 8 show that SCAE is robust to a wide range of $\beta$ values.

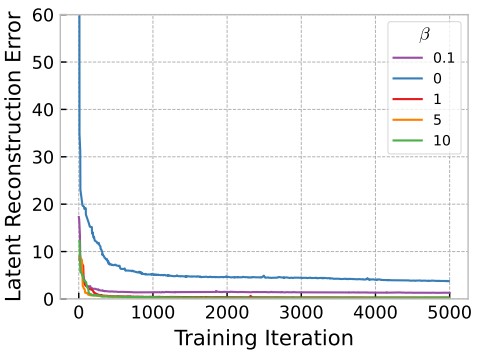 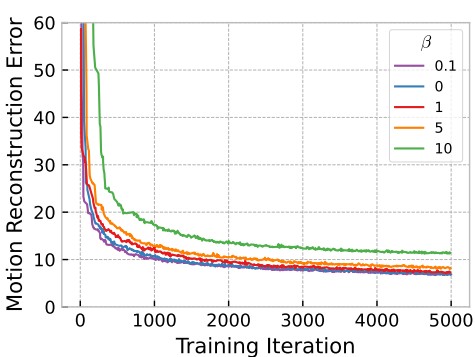

(a) Latent reconstruction error w.r.t. different $\beta$     (b) Motion reconstruction error w.r.t. different $\beta$

Figure 8: The left figure shows that SCAE with small $\beta = 0.1$ can sufficiently improve the latent reconstruction compared to FLD ($\beta = 0$). The right figure shows that only when $\beta = 10$, the motion reconstruction error is slightly increased. In general, when $\beta \sim (0.1 - 5)$, SCAE demonstrates similar motion reconstruction as FLD.

#### A.4.2 More Results on Latent Dynamics Model Learning

Figure 9 compares the learned latent phases across all the 13 motions with different methods. We observe that our method SCAE consistently achieves sparser representations than FLD with fewer frequency components and lower amplitudes.

SCAE learns sparse and well-shaped latent representations. Nonetheless, it retains accurate motion reconstruction as FLD. As shown in Figure 10, both FLD and SCAE accurately reconstruct the motions, which is also validated by the training loss in Figure 3b.

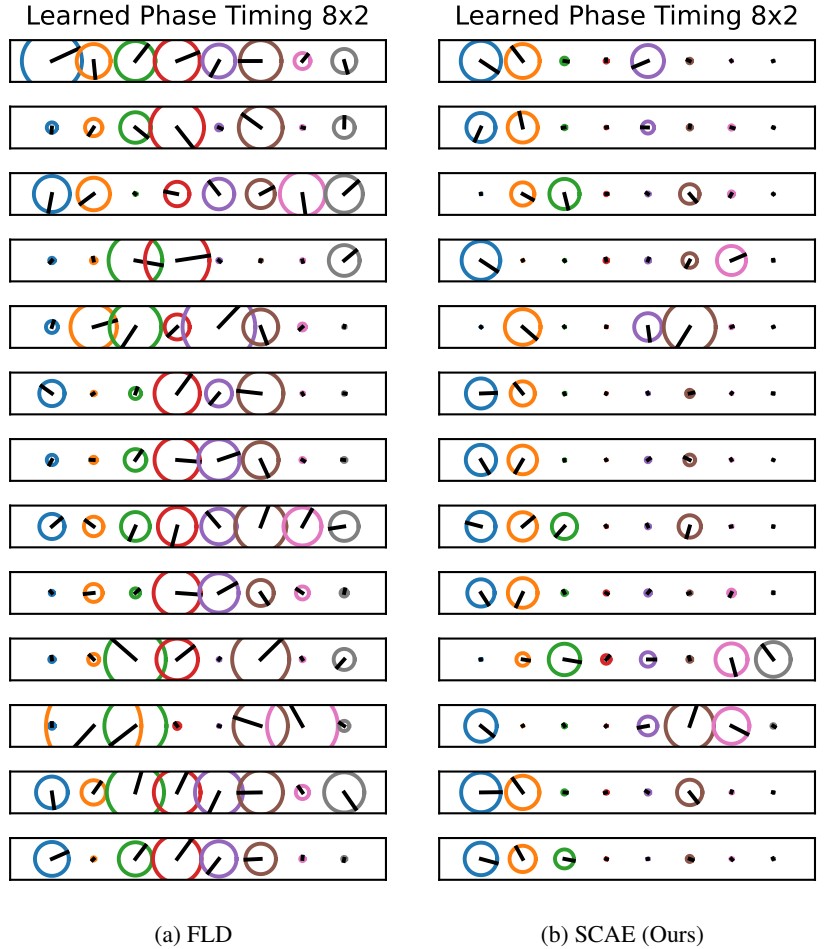

(a) FLD                          (b) SCAE (Ours)

Figure 9: Learned latent phases of 13 different motions. From top to bottom, the motions are: *run, jog, step fast, jump, spin-kick, back, side left, jog slow, side right, cross-over, kick, stride, step*.

### A.4.3    Visualization of Learned Policy

We visualize the motions learned by BMI. In addition to normal motions, such as *stride* in Figure 11a, which can be effectively learned by FLD, BMI successfully acquires two challenging motions *kick* and *jump* in which FLD fails. Figure 12a shows that BMI policy can naturally lift the kicking foot while maintaining the stability of the standing foot. Similarly, Figure 12b illustrates that the robot successfully jumps, with both feet leaving the ground.

However, we note that our policy still struggles with the difficult *spin-kick* and *cross-over* motions which are highly dynamic and can significantly influence the robot balance. Consequently, the robot prioritizes maintaining balance over replicating these motion patterns. For example, the robot rarely lifts its kicking foot in *spin-kick*, and the legs do not fully cross in *cross-over*, as shown in Figure 13.

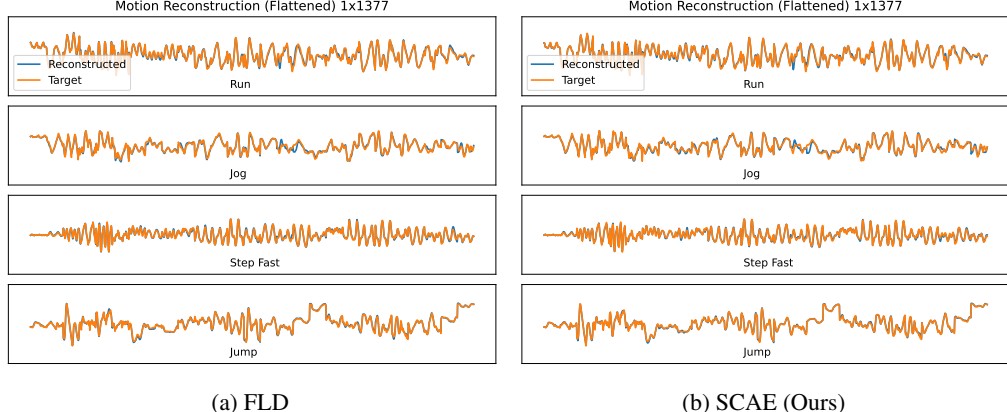

(a) FLD

(b) SCAE (Ours)

Figure 10: Motion reconstruction performance.

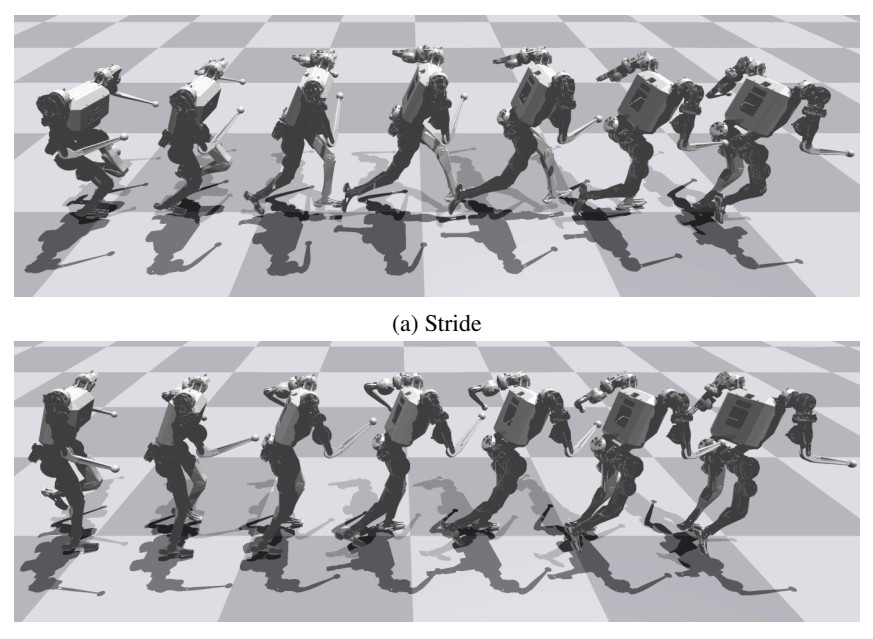

(a) Stride

(b) Back

Figure 11: Normal motions learned by BMI.

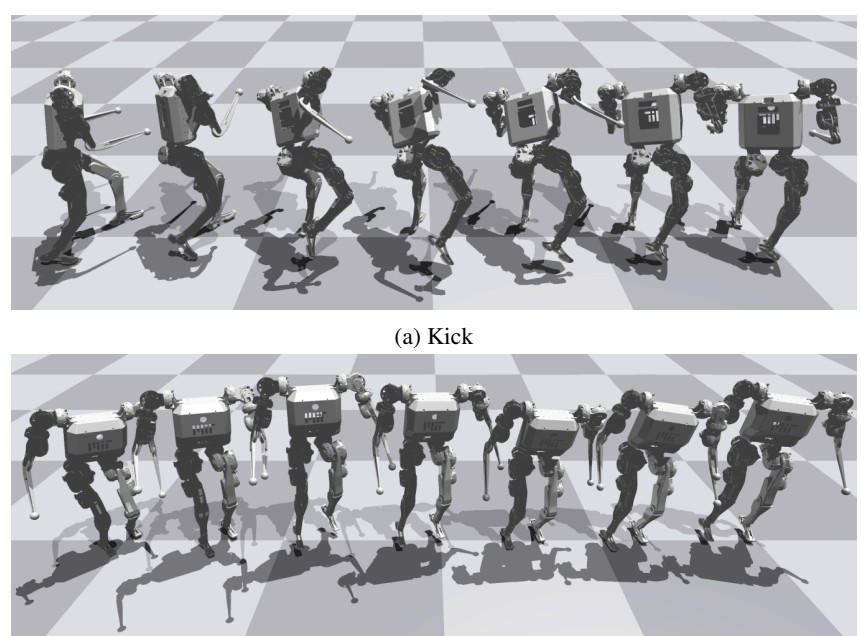

(a) Kick

(b) Jump

Figure 12: Challenging motions learned by BMI.

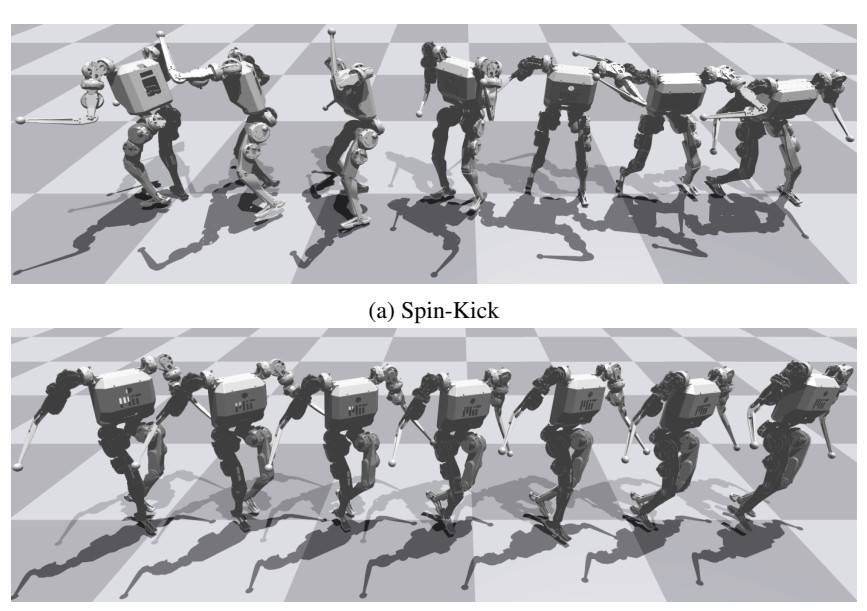

(a) Spin-Kick

(b) Cross-Over

Figure 13: Unsatisfying motions learned by BMI.

