# OpenReview forum: "Bi-Level Motion Imitation for Humanoid Robots"
_robot-learning.org/CoRL/2024/Conference — CoRL 2024_

### Official Review · Reviewer_p8L1 · 2024-07-11
**The quality of the paper, especially the experimental section, seems low.**

**Originality:** 3
**Technical Quality:** 2
**Clarity Of Presentation:** 2
**Potential Impact:** 3
**Recommendation:** 3
**Confidence:** 3

**Review:**

Strengths:
* The paper, especially the proposed method section, is easy to understand.

Weaknesses:
* The benefits of using latent dynamics are not shown experimentally. I would like to see a comparison between the policies of the proposed method and those learned directly from motion capture data without latent dynamics.
* Table 1 and Figure 6 show a comparison of the imitation performance of the conventional and proposed methods, but it is unclear how important the differences in values are.
* It is unclear why a change in the values in Table 1 and Figure 6 can be interpreted as an improvement in imitation performance.
* I would like to see a comparison of the changes in the reference motions from human motion capture data due to the bilevel fine tuning.
* The experiments were conducted only in simulation, and the applicability of the proposed method to real robots is not discussed.
* The accompanying video supposedly shows learned motions such as run, jog, step fast, etc., but it is not clear which motions are performed by the simulated humanoid robot.

**Quality Of The Limitations Section:**

1

**Questions For Rebuttal:**

Issues that need to be addressed by the authors are listed in Weaknesses in the Review section.

**Robotics Focus:**

3

**Summary Of Paper:**

This paper proposed a learning algorithm for self-consistent latent dynamics and an imitation learning method for bilevel fine tuning of policies and latent dynamics. Through an imitation task of 13 different motions of a humanoid robot in simulation, the authors successfully showed that sparse latent phases can be learned by the proposed method and that the latent manifolds are well-formed. The authors also claim that the proposed method improves imitation performance compared to a conventional imitation learning method.

**Summary Of Recommendation:**

The contribution of the paper is not sufficient: weakness in both quality and clarity of the paper.

---

### Official Review · Reviewer_k7pZ · 2024-07-19

**Originality:** 3
**Technical Quality:** 3
**Clarity Of Presentation:** 2
**Potential Impact:** 3
**Recommendation:** 3
**Confidence:** 3

**Review:**

# Strengths

1. The paper introduces a good framework in humanoid robot learning/locomotion, to learn better locomotion skills from the reference motion datasets. The bi-level optimization stage seems to be new and novel.

2. The authors provide a detailed explanation of the SCAE and its role in learning sparse and structured motion representations. The mathematical formulation and the explanation of the bi-level optimization process are clear and well-articulated.


# Weaknesses

1. The proposed method is somewhat incremental to FLD, and the new stages such as bi-level optimization seem to be something common in the learning community, such as finetuing the networks. (however, I am not an expert in this field so I am not very sure.)
2. While the paper explains the methodology clearly, the complexity of the bi-level optimization process might be a barrier for some readers. A more simplified explanation or a high-level overview early in the paper could help in understanding.
3. Figure 2 might need to be largely improved or simplified for better illustrations.
4. Lack of real robot experiments. This might be fine since humanoid robots are expensive, but maybe adding some experiments, such as Sim-to-Sim, to show its applicability, is necessary.

**Quality Of The Limitations Section:**

3

**Questions For Rebuttal:**

Please see weaknesses.

**Robotics Focus:**

3

**Summary Of Paper:**

The paper presents a framework for humanoid robot locomotion, to address the challenge of incorporating physically infeasible motion capture (MoCap) data into training datasets. The proposed method consists of three stages: training an auto-encoder, pre-training a policy, and fine-tuning the decoder and the policy. The proposed method named Bi-Level Motion Imitation (BMI) shows better physically feasible motions in simulation environments.

**Summary Of Recommendation:**

I would recommend reject based on the current techniqual depth and presentations.

---

### Official Review · Reviewer_y7N1 · 2024-07-22
**Bi-Level Motion Imitation for Humanoid Robots**

**Originality:** 3
**Technical Quality:** 3
**Clarity Of Presentation:** 4
**Potential Impact:** 3
**Recommendation:** 3
**Confidence:** 4

**Review:**

Strengths:
- This paper solves the feasibility check of humanoid motion imitation by a bi-level optimziation by learning the latent motion representation and policy together.
- The representation learning design is insightful and inspiring.

Weaknesses:
- Figure 1 needs more detailed caption to guide readers into the data flow and training pipelines.
- The state space and task objectives of motion tracking are not introduced. For example, what are the components of s_t? What rewards do you design for  the rewards in FIgure 2, and what is a_t, is a_t torque or PD setpoints?
- Motion tracking based on latent space is great. I agree with the advantages that the authors claim. But it also comes with several sacrifices. For example, what if you want the robot to walking left and waving the left hand, what latent space should you feed to the policy network?
- The biggest concern I hold for this paper is scalability since the authors only validate the motion representation learning on only 13 motions. What would be the effort and bottleneck of scaling SCAE on the entire AMASS dataset?

**Quality Of The Limitations Section:**

2

**Questions For Rebuttal:**

- I understand "physical feasible motions". But what is the definition of physical consistency? I was confused when reading the introduction.
- "\Tau represents a known time window with H evenly spaced samples". How to get \Tau for each motion sequence?
- Introducing the loss of latent representation computed by feeding the reconstructed trajectory seems tricky to me. Because with a large coefficient \beta, the encoder will just simply converge give the same embedding z for all input motion. The ablation of the \alpha and \beta is needed.
- What is stopping us from seeing a sim2real demo on the MIT humanoid? Is it hardware or RL tuning?
- The metrics in Table is not sufficient enough. More detailed metrics on more diverse motions are needed.


I look forward to discussing with the authors during the rebuttal, and I am open to raise my score if my concerns could be addresed.

**Robotics Focus:**

3

**Summary Of Paper:**

This paper introduces a way to handle infeasible human motions for humanoid learning by a bi-level optimization-based imitation learning framework which alternates between optimizing both the robot policy and the target mocap data.

**Summary Of Recommendation:**

The representation learning design of humanoid learning is novel. My concern is on the scalaibity of this method on 1000x larger motion datasets and sim2real side.

---

### Author Rebuttal · Authors · 2024-08-09

We add a new supplementary material. It includes: (1) updated full paper with appendix; (2) slides; (3) partial videos for the slides. The videos are better shown on the [website](https://sites.google.com/view/bmi-corl2024).

---

### Decision · Program_Chairs · 2024-09-04

**Decision:**

Accept

**Comment:**

This paper introduces an approach to tackle infeasible human motions in humanoid learning using a bi-level optimization-based imitation learning framework. The framework alternates between optimizing the robot policy and the latent dynamics.

Reviewers found the use of latent motion representation and bi-level optimization to be inspiring. However, before the rebuttal, there were concerns about the absence of real-world experiments and the lack of thorough discussions on the limitations and trade-offs of motion tracking based on latent space. Additionally, the paper lacks detailed explanations for some of the figures, equations, and videos.

During the rebuttal, the authors did an excellent job addressing the reviewers’ concerns by including new experiments on sim-to-sim transfer and conducting additional analysis/ablations on the bi-level finetuning and the latent parameterization space. The authors also made substantial improvements to the presentation of the work. As a result, all reviewers now rate the paper as a Weak Accept, and the AC agrees that the paper is now much stronger, with the major limitation being the lack of real-world experiments. However, the AC acknowledges the challenges of conducting hardware experiments within the short rebuttal period and appreciates the inclusion of sim-to-sim transfer results.

To further strengthen the paper, the authors should consider Reviewer y7N1’s suggestion to scale to AMASS and transfer the policy to a higher-fidelity simulator like Mujoco or an actual real robot.